# Designing an Energy-Resilient Neighbourhood Using an Urban Building Energy Model

**Niall Buckley** [1,*], **Gerald Mills** [1], **Samuel Letellier-Duchesne** [2] and **Khadija Benis** [2]

1 School of Geography, University College Dublin, D04 V1W8 Dublin, Ireland; gerald.mills@ucd.ie
2 Sustainable Design Lab, Massachusetts Institute of Technology, Cambridge, MA 02139, USA; samueld@mit.edu (S.L.-D.); kbenis@mit.edu (K.B.)
* Correspondence: niall.buckley1@ucdconnect.ie

**Abstract:** A climate resilient city, perforce, has an efficient and robust energy infrastructure that can harvest local energy resources and match energy sources and sinks that vary over space and time. This paper explores the use of an urban building energy model (UBEM) to examine the potential for creating a near-zero carbon neighbourhood in Dublin (Ireland) that is characterised by diverse land-uses and old and new building stock. UBEMs are a relatively new tool that allows the simulation of building energy demand across an urbanised landscape and can account for building layout, including the effects of overshadowing and the potential for facade retrofits and energy generation. In this research, a novel geographic database of buildings is created using archetypes, and the associated information on dimensions, fabric and energy systems is integrated into the Urban Modelling Interface (UMI). The model is used to simulate current and future energy demand based on climate change projections and to test scenarios that apply retrofits to the existing stock and that link proximate land-uses and land-covers. The latter allows a significant decoupling of the neighbourhood from an offsite electricity generation station with a high carbon output. The findings of this paper demonstrate that treating neighbourhoods as single energy entities rather than collections of individual sectors allows the development of bespoke carbon reducing scenarios that are geographically situated. The work shows the value of a neighbourhood-based approach to energy management using UBEMs.

**Keywords:** UBEM; UMI; zero-carbon; climate resilience; digital twin; renewables; closed loop

## 1. Introduction

Cities have been described as 'entropic black holes' as they are sustained by resources far outside their boundaries [1]. Making cities more sustainable requires increased efficiencies by managing the attributes of the urban system (its physical form and functions) to reduce demand and recycle resources internally, thus closing open links between sources and demand. The latter has been characterised as urban harvesting to create a flexible network of resource linkages that reduces reliance on one resource and enhances resilience. A key resource that links land-cover (form) and land-use (function) is the flux of energy, which is generally attributed to building, transport and industrial sectors [2–4]. While cities are a focus of energy use (and $CO_2$ emissions), they are also ideally scaled for reconfiguring systems of energy production, distribution and reuse owing to the juxtaposition of different functions.

Buildings are responsible for 40% of energy consumption globally, which corresponds to 33% of greenhouse gas (GHG)-related emissions [5]. In the EU, the Green Deal sets out a roadmap to achieve 50% reduction in GHG emissions by 2030 and carbon neutrality by 2050. To meet the 2030 target, the roadmap requires reductions of 14% building final energy consumption, of 18% heating-related energy consumption and of 60% in GHG-related emissions. Many of the policy tools are based on Energy Saving Measures (ESMs), such as retrofits to improve the thermal properties of the building envelope and the efficiency of

HVAC systems. However, ESMs are currently applied to just 1% of the European building stock annually [6,7], and the same is true in Ireland [8,9]. The effectiveness of ESMs is typically assessed using building energy simulations (BESs), which rely on modelling tools that are distinguished by the precision and scale of application. Detailed BESs have relied on building energy models that require considerable information on individual buildings, which can be representative of a category of building type (an archetype). Urban building energy models (UBEMs) are a recent development that permit simulation of multiple buildings using parsimonious information on the building stock. While the results are less precise for individual buildings, they can account for urban layout (and its impact on daylight and shadowing, for example) and allow for evaluation of neighbourhood-scale ESMs, including community retrofits and energy grid management [10]. This research focusses on the application of a UBEM to a diverse neighbourhood in Dublin (Ireland) to explore the best pathway to achieving energy resilience through a combination of building retrofits and energy sharing among proximate urban functions and local resources.

## 2. Materials and Methods

In the field of building energy management there are two mainstream approaches to estimating demand across a building stock, namely top-down and bottom-up [11,12]. Both methods have advantages and drawbacks, and each will be better suited for different applications. For instance, the top-down approach has been traditionally used to estimate energy demand for existing buildings stock in the context of grid-energy management. Statistical methods are employed to disaggregate energy into coarse (e.g., commercial, industrial and residential) and finer (e.g., dwelling and business types) resolution energy use. This method needs few computational resources and low levels of data, yet can be reasonably accurate. However, the top-down approach is mostly static and does not support energy simulations that can test responses to changes in policy or technology or behaviour. Alternatively, bottom-up approaches can capture the complexities of heat flow within and around buildings and the influence other buildings and objects have on energy demand [13]. These methods can simulate the impact of ESMs as building fabric, systems and occupant patterns can be altered to describe future scenarios. These models can be used to calculate energy use for representative building profiles (archetypes) using weather files that are aggregated to match the building stock at urban, regional or national levels [14–16]. Until recently, the data resolution and computational power needed for bottom-up approaches has been a major obstacle, but advancements in the field of BES software has made them more feasible for use [10,13].

Building energy models (BEMs) are the standard tool for understanding and simulating energy use and losses and for testing management strategies. BEMs have evolved from simple steady-state to transient models that can capture dynamic conditions in the real world [17,18]. However, applying these highly complex BEMs to examine the energy exchanges within and between many neighbouring buildings simultaneously is not feasible yet. UBEMs have emerged as tools for examining energy use in neighbourhoods. To run efficiently, these UBEMs must compromise on the precision of simulation for an individual building, but they can evaluate the impacts of ESMs that change the form and function of neighbourhoods with a reasonable level of accuracy [13,19,20]. Moreover, these models can address issues of energy harvesting and sharing at a community scale using novel technologies, which are beyond the scope of BEMs.

The range of building types in a neighbourhood coupled with the varying occupant profiles and seasonal variation in energy needs represent major simulation challenges and opportunities for UBEMs. For example, in a seasonal climate dominated by heating needs, significant solar energy potential may exist in summer months when there is little demand for space heating, which means energy storage systems (ESS) may be needed, but these are expensive and cannot store energy for long periods of time [21]. However, urban areas have a diverse range of energy consuming activities that are independent of seasonal variation. Energy credit schemes can be used to distribute excess energy during off-peak demand

among users to reduce the overall energy demand by a neighbourhood [22]. UBEMs have the capability of modelling temporal energy yields onsite to meet or mitigate temporal energy demand [23]. This type of modelling provides the opportunity to examine both neighbourhood energy grid management (e.g., energy transfers between proximate users with different temporal needs) and building retrofits [24]. UBEMs have been used to trial ESM policy goals to estimate the time and cost associated with hitting certain targets [25,26] and to assess how best to address energy poverty [27,28]. Finally, UBEMs can be linked to urban systems models that consider wider energy-related socio-economic issues such work–home commuting patterns, district heating and local food production [23,29,30]. Apart from the computational resources required to run a UBEM for a diverse neighbourhood, the main obstacle for their wider use is the need for detailed geospatial information on the building stock and uses.

Resilience in cities has been defined as a city's ability for both its socio-technical and ecological systems scales, to adapt and/or maintain functions when faced with disturbances [31,32]. Urban energy harvesting seeks to take advantage of local supply and demand management to reduce reliance on external resources, which are often carbon intensive. To accomplish this, it is important to map its magnitude, quality and temporal characteristics [32]. This paper applies a UBEM—the Urban Modelling Interface (UMI)—to examine the energy profile of a neighbourhood that is experiencing significant redevelopment as tall apartment blocks are inserted into an urban landscape with commercial offices and a variety of residential buildings, many of which have poor energy ratings. UMI simulates building- and neighbourhood-scale energy demand using building archetype data that is linked to a geographic database. Here, these data are derived from a GIS database created using an EU building categorisation database that provides basic information on building types. UMI is used to explore pathways toward carbon neutrality in line with the EU Green Deal, using the example of a neighbourhood in Dublin (Ireland).

The structure of the paper outlines the methodology (the UMI infrastructure and methods and the study area and its current and future building make-up), the results of simulations for the baseline (current) situation and future outcomes based on the implementation of ESMs and projected climate change. We consider the potential for energy sharing within the neighbourhood using solar power and district heating/cooling systems [30]. We also consider the impact of novel carbon mitigating interventions, such as closed environment agriculture [29].

## 3. Methodology

The first section of the methodology introduces how UMI operates and its data requirements, and briefly details how these data are generated. The second part of the methodology outlines the study area and how UMI simulations help examine the potential for creating a near-zero carbon neighbourhood in Dublin.

### 3.1. Urban Modelling Interface (UMI)

UMI is an urban building energy modelling platform that uses EnergyPlus at its core to simulate energy use for multiple buildings that make-up extensive urban areas comprised of diverse user needs and occupation patterns [23]. UMI can account for mutual interactions among buildings including shadowing and reflections. Each building is divided into core and conditioned parts; the latter form the occupied space at the perimeter of the building envelope and are bounded on one side by the façade. Detailed simulations are performed on the conditioned parts. To overcome the computation demands associated with simulating the energy uses of large groups of buildings, it uses a sampling technique (Shoeboxer) to select representative conditioned zones from buildings. This is done using an annual solar radiation map of the building facades that are clustered into groups of similar radiation intensity [19]. The results from the sample are applied to the population of buildings under study. While this procedure reduces the precision of the findings for individual buildings, it provides sufficiently accurate results for neighbourhoods rapidly.

The UMI integrates geospatially referenced building data that describes the fabric of the envelope, the building energy systems and the typical occupation patterns. The model permits the examination of ESMs that assess the consequences of change to building materials, upgrade heating/cooling systems, integration of PV generated solar energy and so on. In addition, it can incorporate to neighbourhood-scale changes, such as energy redistribution and home–work commutes. The critical information to run these models is the provision of detailed neighbourhood-scale building data.

The geospatial database used in this work was created by linking building archetypes and their attributes to a GIS database of building footprints. The residential archetypes were generated as part of an EU project (Tabula/Episcope) that categorised the national building stock into types, each of which is linked to descriptions of the original construction properties (walls, roof and windows) and heating systems. The project allows for rapid assessment of the potential of retrofits and expresses the energy demand in terms of annual energy use intensity (EUI in kWh m$^2$). These residential archetypes and their data must be translated into templates to be incorporated into UMI, but those associated with commercial buildings are available within the software (based on SIA archetypes and DOE occupant schedules). Once created, the templates for all buildings can be modified to simulate ESMs to reduce energy demand (and carbon emissions); these measures include changes to the fabric (e.g., insulation and glazing), to heating/cooling systems (both technological and fuel sources) and occupancy schedules and set temperatures [33–35]. In this study, all the buildings in Dublin city centre (about 30,000) were classified into these archetypes using imagery (GoogleStreetview), census data (which indicates age of dwelling) and field work; this process is described in detail elsewhere [36]. Data from the Tabula archetype tables was used to construct 3D building templates using Rhino and to generate the EnergyPlus files used by UMI. These steps are described in Buckley et al. [25], which also evaluated the simulations of residential EUI values reported in Tabula and against BER data. The entire process creates a digital 'twin' of the study area with capability of modelling spatiotemporal energy flows.

### 3.2. Neighbourhood Study Area

The study area located in Dublin city centre and was chosen based on its relative isolation, mix of residential buildings with diverse socio-economic profiles, adjacency to a transportation hub and a planned large mixed-use development (Figure 1). The boundary of the study area was the river Liffey (south-side), the Royal canal (east side) and the railway lines (west side), which form a triangular area of 0.5 km$^2$. The landscape here was dominated by port activities until the middle of the 20th century, and many of the buildings in this area were large warehouses used for storage of goods that were being transferred to/from the railway and canal system. Since the 1970s port activities have declined, the canals and the riverside docks are unused and the train system now handles mostly passenger transport. Since the 1990s the area has developed as a financial services centre, and the new buildings in this area are comprised of large commercial office space and apartment buildings.

The current building composition of the study area (Table 1) consists of residences (a mix of both single-family homes (SFHs) and apartment blocks (ABs)) and modern office blocks. Much of the SFHs were built before the 1980s and the start of building energy standards; a significant proportion of these consists of solid brick terraced houses constructed before 1950. Some SFHs were constructed in the 1990s, but the bulk of modern buildings are apartment blocks (ABs) constructed after 1990. The 2016 household census provides some detail on the socio-demographic composition of the census areas that make-up the neighbourhood. The Deprivation Index (DI, or socioeconomic status) is based on the demographic profile, social class composition and labour market situation for each area; scores indicate position relative to the average socioeconomic status, and higher negative scores indicate higher levels of deprivation and vice versa) (Haase T. and Pratschke J. (2017) The 2016 Pobal HP Deprivation Index for Small Areas (SA): Introduction and

Reference Tables. Available at: https://www.pobal.ie/app/uploads/2018/06/The-2016
-Pobal-HP-Deprivation-Index-Introduction-07.pdf) (accessed on 1 April 2021). There were
3061 residents in 2016 of which 43% lived in single-family homes (SFHs) and the remainder
in apartment blocks (ABs). Most of the families (74%) living in SFHs were classed as semi-
skilled/unskilled, while 87% of adults in ABs were classed as professional/manager; 12%
of those living in SFHs were renters, while 88% of those in apartments were renters. The
office buildings were constructed since the 1990s and are mostly associated with financial
service functions.

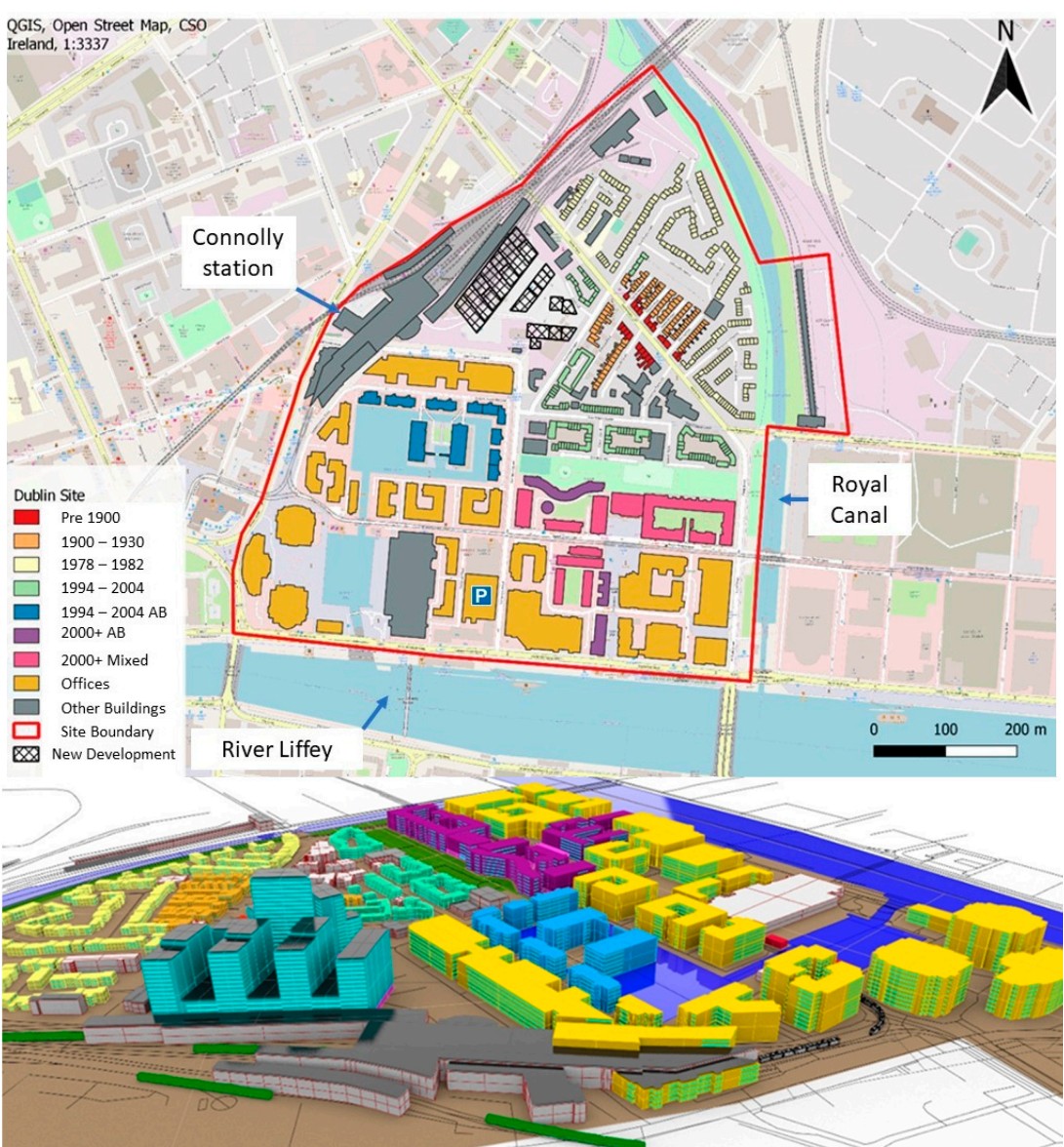

**Figure 1.** A map and 3D rendering of the neighbourhood study area.

Building Energy Ratings (BERs; generated as part of an energy performance certificate
(EPC)) are required of every dwelling sold or rented since 2008; the BER grade is based on
an evaluation of the dwelling following an inspection that generates an EUI (kWhm$^{-2}$).
The EUI assigned depends on both the building fabric and the heating system, including
the energy source. The reported BER values are an imperfect sample of the building
stock, much of which has undergone no envelope retrofits since built, although many
will be heated by gas and/or electricity replacing the original solid fuel fireplaces. Where
the supply system is gas/oil, the efficiency of the heating system in the dwelling (or

building) is key, and the typical coefficient of performance for a conventional system from the 1980s onwards is between 0.6 and 0.8. Where the energy supply is electricity, the BER assessment accounts for the efficiency of the generation system, which is offsite; consequently, electricity-based heating systems are relatively inefficient compared to gas-based system. These differences partly explain the variation in BER ratings within the study area shown in Table 1. The overall conclusion is that more deprived populations generally occupy older houses with poorer energy standards.

**Table 1.** A list of the census areas within the study area, alongside information based on the 2016 household census, the age of housing from building archetype classification and BER (https://www.seai.ie/publications/Your-Guide-to-Building-Energy-Rating.pdf accessed on 1 April 2021). The census information identifies the relative deprivation score, and the population in single family homes (SFH) and apartment blocks (AB). The age of buildings is categorised by age of construction. The BER information is based on surveys that are required of dwellings sold or rented (there are many dwellings within an apartment block).

| Dublin Site Small Areas Numeric IDs | Residential Population | | | Age of Housing (N) | | | | BER | | BER Samples (Surveys) |
|---|---|---|---|---|---|---|---|---|---|---|
| | DI | SFH | AB | Pre–1978 | 1978–1982 | 1983–1999 | 2000+ | Median kWhm$^{-2}$ (BER Grade) | | |
| 268109001 | −26.69 | 342 | 8 | 0 | 100 | 4 | 0 | 192 | (C2) | 85 |
| 268109002 | −21.18 | 253 | 0 | 0 | 87 | 0 | 0 | 213 | (C3) | 78 |
| 268109003 | −14.74 | 140 | 1 | 85 | 0 | 0 | 0 | 398 | (F) | 43 |
| 268109004 | −14.96 | 175 | 46 | 32 | 0 | 27 | 0 | 283 | (D2) | 55 |
| 268109005 | −22.10 | 214 | 92 | 0 | 0 | 62 | 0 | 130 | (B2) | 252 |
| 268109006 | 13.28 | 0 | 103 | 0 | 0 | 0 | 1 | 209 | (C3) | 46 |
| 268109007 | 19.49 | 0 | 169 | 0 | 0 | 0 | 3 | 188 | (C2) | 66 |
| 268109008 | 19.06 | 0 | 223 | 0 | 0 | 2 | 0 | 248 | (D1) | 49 |
| 268109009 | 23.59 | 0 | 375 | 0 | 0 | 0 | 2 | 176 | (C2) | 57 |
| 268109010 | 22.36 | 0 | 297 | 0 | 0 | 0 | 1 | 160 | (C1) | 135 |
| 268109011 | 27.41 | 0 | 218 | 0 | 0 | 0 | 2 | 165 | (C1) | 111 |
| 268109012 | 15.65 | 0 | 272 | 0 | 0 | 0 | 2 | 178 | (C2) | 37 |
| 268109013 | 23.42 | 0 | 211 | 0 | 0 | 6 | 0 | 226 | (D1) | 73 |
| 268109014 | 19.79 | 0 | 135 | 0 | 0 | 2 | 0 | 234 | (D1) | 49 |
| Area | | 1124 | 2150 | 117 | 187 | 103 | 11 | | | |

The new development (Connolly Quarter) planned for this neighbourhood (Figure 1) is a mixed-use high-density site consisting of residential properties with some retail. The buildings are arranged into blocks 4 to 23 storeys in height on an elevated site currently used for car parking. It will have 741 residential units and supporting amenities, such as green space. This development will increase the population by over 2900 (nearly doubling the population of the area) and increase the floor space significantly. However, as the anticipated EUI is <25 kWhm$^{-2}$ (BER of A1), it will also reduce the overall EUI of the neighbourhood. The new development is located next to a major transportation hub (Connolly station) that includes a bus depot, a regional train station and an urban light rail system (LUAS). This spatial juxtaposition is important, as the station canopy provides an extensive area for PV installation, and the LUAS is run on electricity (demand 360 mWh per month) from the national grid [37].

*3.3. UMI Simulations*

A series of simulations was performed on the building stock by comparing the energy performance of the neighbourhood (assuming no change from the original build) in the current climate and projected 2050 climate; examples of residential building stock are provided in Figure 2. The focus of this work is on residential single-family houses (SFHs) and apartment blocks (ABs), but the energy demand of office buildings was taken into account when we considered the potential for energy harvesting and sharing. The new residential development was included in the UMI model as an additional 108,139 m$^2$ of available floor space. Following the application of standard retrofits appropriate to the types of residential building present, these current and future climate simulations were repeated. Finally, we considered the value of treating the entire neighbourhood as a single entity in which energy is captured and shared among residential units to match demand profiles. In all simulations, the energy supply used the same heating system.

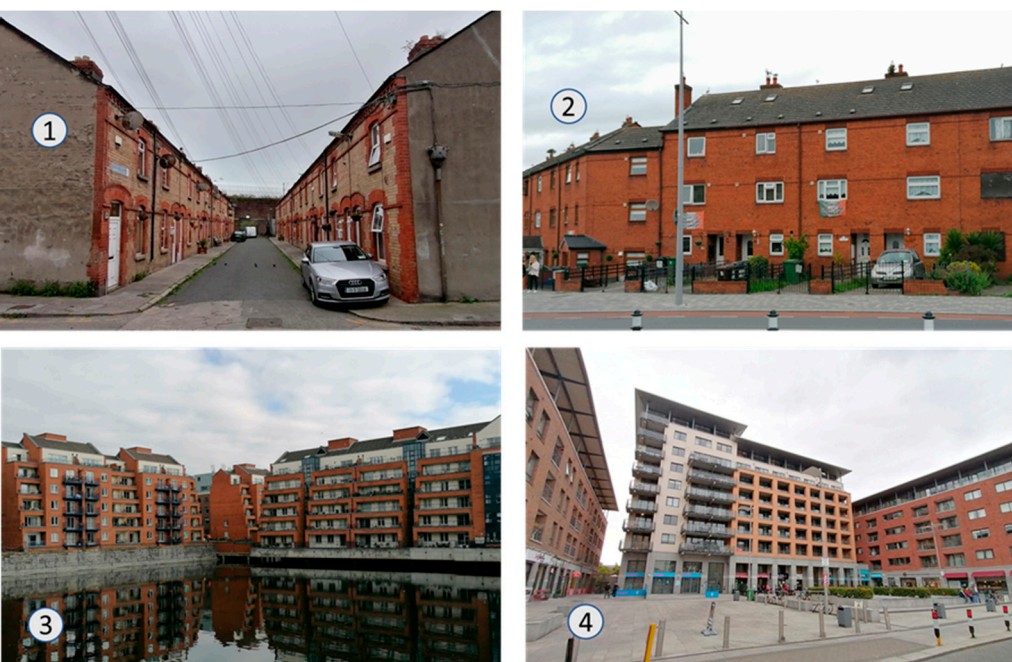

**Figure 2.** Residential building types in the study area: single family homes constructed in terraced layout prior to 1980 (**1**) and between 1980 and 2000 (**2**), and apartment blocks constructed in the 1990s (**3**) and after 2000 (**4**).

The standard Dublin EPW weather file available from DOE EnergyPlus website was used for current climate, which is based on observations made at Dublin Airport (5 km from the study area). These simulations provided the baseline results against which the impacts of ESMs and climate change were evaluated. The Climate Change World Weather File Generator for World-Wide Weather Data (CCworldWeatherGen) [38,39] was used for future climate conditions. This process altered the Dublin EPW file in line with projections under an IPCC emissions scenario (A2), for which expected global $CO_2$ concentrations for 2050 are 575 ppm.

Here, this approach was simply used to indicate the likely change in background climate conditions to judge how the current residential building stock will be impacted by a warmer climate. Figure 3 shows monthly heating degree days (HDD) using a base temperature of 15.5 °C, which suggests that the annual HDD will decrease from 2166 to 1751 by 2050; by comparison, the annual cooling degree days using a base temperature of 22 °C increases from 1 to 3 days. Thus, we may expect that the impact of climate change over the next 30 years will be to reduce the annual residential heating demand and have a marginal impact on cooling demand.

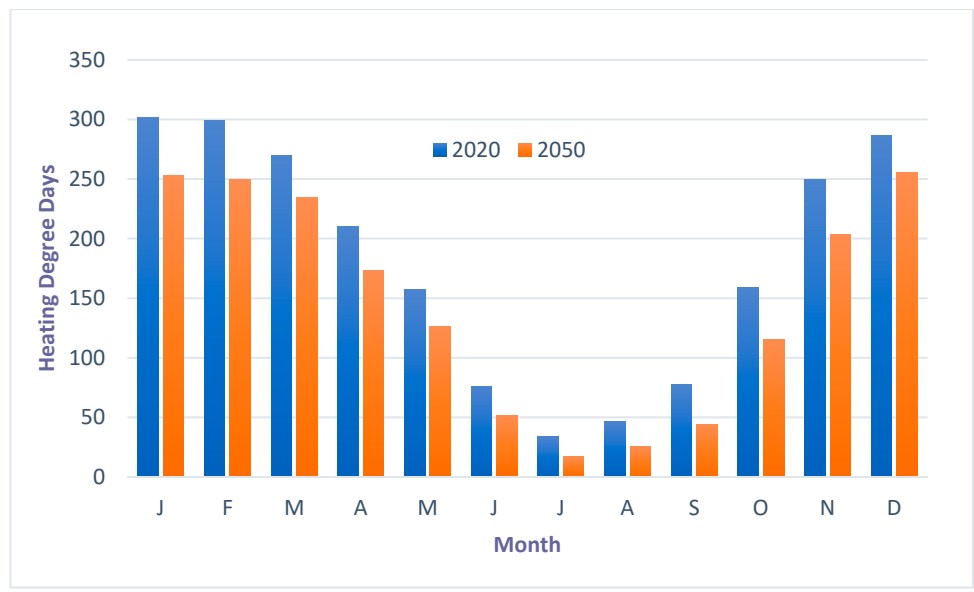

**Figure 3.** Heating degree days using the current Dublin weather file (2020) and the modified file to represent the impact of climate change (2050).

To place the results in context, it is worth examining energy (heating and cooling) demand in the neighbourhood overall. Figure 4 shows that the bulk of the heating demand currently is driven by residential needs and that the offices have both heating and cooling demands, owing to the considerable internal energy generation during daytime work hours. The potential solar energy generation within the neighbourhood is also shown on this graph.

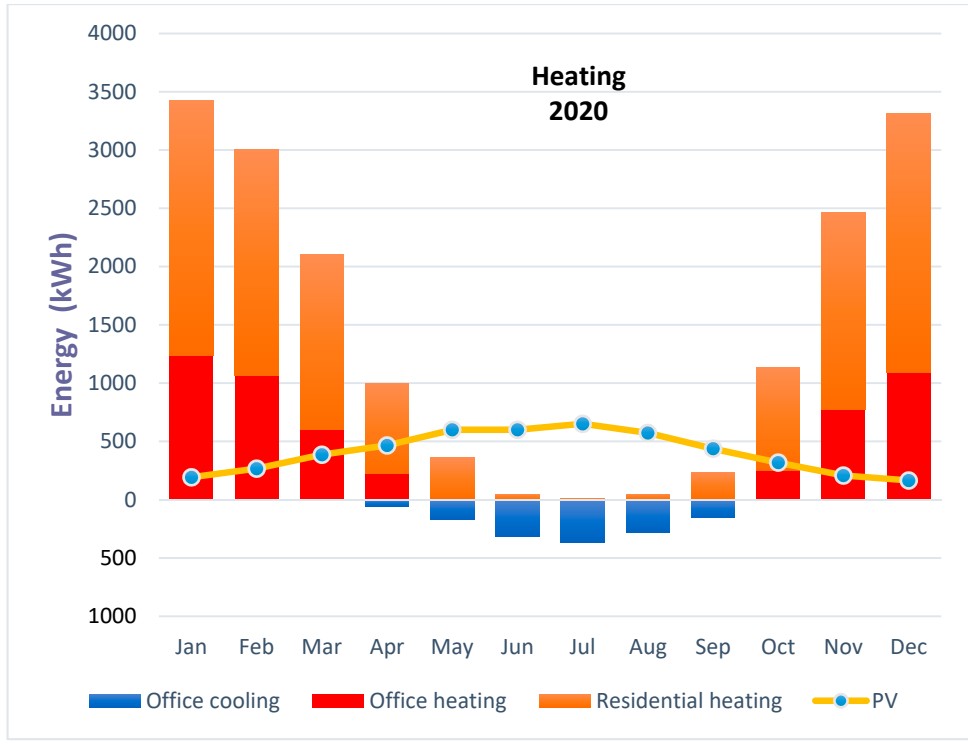

**Figure 4.** Simulated heating (positive) and cooling (negative) energy demand in the neighbourhood for office and residential buildings. The energy generated by photovoltaic (PV) surfaces using the rooftops of public buildings in the neighbourhood is also shown.

## 4. Results

### 4.1. Status Quo (SQ)

Table 2 shows the different EUI values associated with each of the building types assuming no change to the envelope since built (that is, status quo). The residential building simulations were evaluated against reported EUI and showed good agreement; these comparisons are discussed elsewhere [25]. In this study, office buildings were benchmarked against CIBSE's guide F, which estimates good practice air-conditioned office buildings to have an EUI between 97 and 114 kWh/m$^2$/year. Keep in mind that we assumed the same energy supply (and COP) across the residential neighbourhoods, so that the focus here was on the building fabric only. The predicted EUI for the SFH were 313 and 203 kWhm$^{-2}$ for the older and newer buildings and, for apartment blocks (AB), 147 and 140 kWhm$^{-2}$ for the older and newer buildings, respectively. The new development (ND) had an EUI (25 kWhm$^{-2}$) that was significantly lower than the existing stock, which averaged 183 kWhm$^{-2}$. Adding the ND alone would reduce the average EUI for the neighbourhood to 115 kWhm$^{-2}$ and the per capita energy use and associated carbon emissions from 1.87 to 1.13 tCO$_2$ (Table 2). If the neighbourhood were to remain as is until 2050, then the impact of warming temperatures would be to reduce the heating demand while having a negligible impact on cooling demand. The neighbourhood averaged EUI with the ND would now be 100 kWhm$^{-2}$, which is 45% lower than the 2020 EUI without the ND.

**Table 2.** Summary of UMI simulation results for the residential sector in the study area. The CO$_2$ emissions are estimated using 0.237 kg CO$_2$ (kWh)$^{-1}$ based on the fuel mix in Ireland.

| | SFH Pre–1980 | SFH 1981–1999 | AB 1981–1999 | AB 2000+ | ND | Offices | Existing Residential (ND) |
|---|---|---|---|---|---|---|---|
| Conditioned Zone (m$^2$) | 31,488 | 16,954 | 31,469 | 75,403 | 118,125 | 374,519 | 155,314 (273,439) |
| Population | 600 | 350 | 782 | 1879 | 2940 | 20,589 | 3611 (6551) |
| Status Quo | | | | | | | |
| 2020 EUI (kWhm$^{-2}$) | 312.9 | 202.9 | 147.0 | 139.6 | 24.6 | 99 | 183 (115) |
| t CO$_2$ per capita | 3.89 | 2.33 | 1.40 | 1.33 | 0.23 | 0.59 | 1.87 (1.13) |
| 2050 EUI (kWhm$^{-2}$) | 255.1 | 173.0 | 131.4 | 125.6 | 22.3 | 95 | 158 (100) |
| Standard retrofits | | | | | | | |
| 2020 EUI (kWhm$^{-2}$) | 23.0 | 22.1 | 21.7 | 23.3 | 24.6 | 99 | 21 (23) |
| t CO$_2$ per capita | 0.29 | 0.25 | 0.21 | 0.22 | 0.23 | 0.59 | 0.23 |
| 2050 EUI (kWhm$^{-2}$) | 20.9 | 20.6 | 20.6 | 21.8 | 22.3 | 95 | 20 (22) |

### 4.2. Retrofit (R)

Table 2 shows the simulations following a complete retrofit to the existing building stock that increased the U values of the envelope by insulating the roof and walls, replacing the windows, reducing heat loss by infiltration and replacing the lighting system. The default and retrofit parameter values used in the UMI simulation are shown in Table 3. The Coefficient of Performance (CoP) describes the efficiency of the heating system; in the simulations we replaced conventional systems that have CoP values < 1 with heat pumps that are more than 5 times more efficient. Table 3 also shows estimated cost per dwelling that accounts for the floor size and the nature of the envelope, especially the walls. Buildings constructed prior to 1950 have solid wall construction, while those constructed since use cavity walls that are easier to insulate. Based on the conditioned areas (Table 1), the cost to upgrade the existing residential stock was estimated at about €35 million for the materials. Currently, retrofits are applied on a building-by-building basis; financial incentives by the state to retrofit are available to homeowners who must pay for the

refurbishment and then claim the associated grant. However, this support system process has resulted in a low take-up that leaves significant parts of the energy efficient stock untouched.

**Table 3.** Parameter values for components of buildings before status quo (SQ) and after retrofit (R) modifications and estimated cost per dwelling based on size. The default system used to heat the dwelling is a heat pump. For solid wall buildings and apartments, external wall insulation is applied. All costs and associated EUIs for each measure are derived from the Tabula webtool and associated literature linked below (https://episcope.eu/fileadmin/tabula/public/docs/brochure/IE_TABULA_TypologyBrochure_EnergyAction.pdf) (accessed on 20 February 2021).

| | SFH Pre–1980 | | | | | | SFH 1981–1999 | | AB 1981–1999 | | AB 2000+ | |
| | Georgian | | Solid Brick | | Cavity | | | | | | | |
| Component | SQ | R | SQ | R | SQ | R | SQ | R | SQ | R | SQ | R |
|---|---|---|---|---|---|---|---|---|---|---|---|---|
| Roof (U) $W(m^2K)^{-1}$ | 1.35 | 0.13 | 0.68 | 0.13 | 0.4 | 0.13 | 0.26 | 0.13 | 0.4 | 0.15 | 0.35 | 0.14 |
| Façade (U) $W(m^2K)^{-1}$ | 1.5 | 0.27 | 2.1 | 0.24 | 1.1 | 0.41 | 0.55 | 0.32 | 0.6 | 0.16 | 0.55 | 0.15 |
| Floor (U) $W(m^2K)^{-1}$ | 0.61 | 0.61 | 1.58 | 1.58 | 0.86 | - | 0.66 | - | 1.22 | - | 0.86 | - |
| Windows (U) $W(m^2K)^{-1}$ | 4.8 | 2 | 5.7 | 1.3 | 3.7 | 1.3 | 2.8 | - | 4.8 | 1.3 | 2.8 | 1.3 |
| System (CoP) | 0.62 | 5 | 0.62 | 5 | 0.62 | 5 | 0.62 | 5 | 0.62 | 5 | 0.62 | 5 |
| Lighting ($Wm^{-2}$) | 5 | 1.8 | 5 | 1.8 | 5 | 1.8 | 5 | 1.8 | 5 | 1.8 | 5 | 1.8 |
| Air tightness (ACH) | 0.4 | 0.2 | 0.4 | 0.1 | 0.2 | 0.1 | 0.2 | 0.05 | 0.2 | - | 0.2 | - |
| Cost per dwelling | €47,000 | | €37,000 | | €22,000 | | €19,000 | | €15,000 | | €15,000 | |
| Cost per m² | €265 | | €231 | | €211 | | €117 | | €239 | | €239 | |

### 4.3. Energy Harvesting and Distribution

A retrofit programme, such as that described above, would increase the energy efficiency of every dwelling; in this case, the database that underpins the UBEM identifies the clusters of buildings in an area that may benefit from an economies of scale approach to retrofits. A holistic approach would treat the entire area as a single energy entity and consider how best to harvest and redistribute energy within the neighbourhood based on the temporal and spatial supply and demand. This approach enhances energy resilience by reducing the reliance on a single supply of energy and linking users at a community scale. The advantage of an UBEM is that it allows us to test a variety of energy scenarios designed around the local geography of resources. Sources used for estimating cost-, energy- and $CO_2$-related savings are listed in the Appendix A of this paper.

In the neighbourhood case-study, most of the energy supply is obtained from the national grid, which currently has a carbon intensity of 324 $gCO_2$ per kWh, which has been falling due to fuel switches and incorporation of wind power (https://www.seai.ie/publications/Energy-Emissions-Report-2020.pdf) (accessed on 20 February 2021). Figure 5 shows a diagram of the neighbourhood, which identifies the main energy sinks (residences, offices and the LUAS) and the local resources. The latter includes extensive flat roof areas that can be used for solar energy generation and water in the canal system that can be integrated into a district heating/cooling system.

The potential roof space for PV installation includes the local railway canopy, the flat-roofed public buildings in the area and the roof of the new development (30,613 m²). If all this space were used, the PV area would cover over 3 hectares and generate a monthly average of 405 MWh, which is concentrated in the daytime summer months when the energy is more than three times (600 MWh) the value in winter. Assuming a cost of €550 $m^{-2}$, the overall cost of this system is just over €17 m (https://www.seai.ie/publications/Best_Practice_Guide_for_PV.pdf) (accessed 21 February 2021). The challenge is how best to use this energy by matching the resource to the demand; there are four options:

1. Export to the national grid.
2. Offset the winter-time residential heating demand.
3. Offset the summer-time office cooling demand.
4. Offset all-year energy demand of Luas.

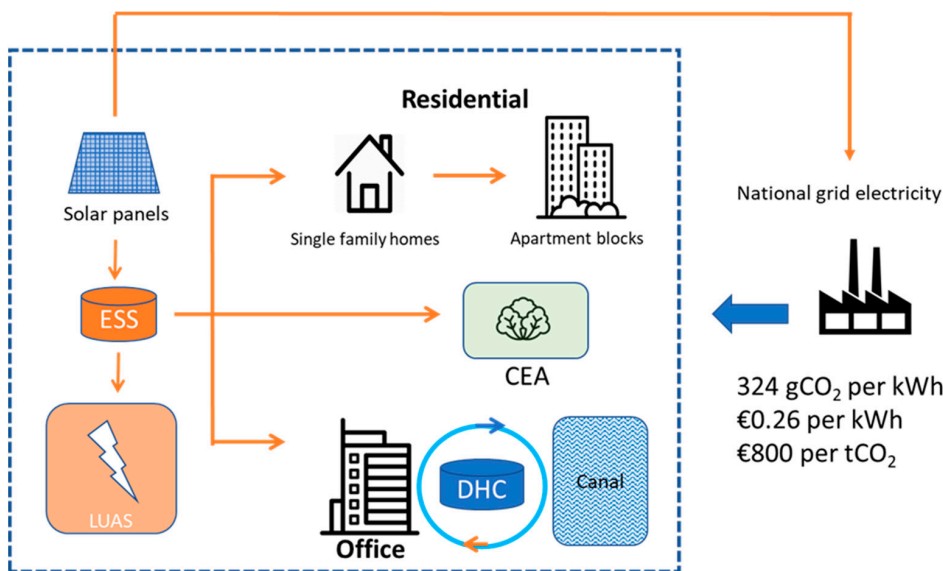

**Figure 5.** A conceptual diagram of the neighbourhood study area (marked by the dashed box) and the local resources available. Shown here is the Energy Storage System (ESS) and the Controlled Environmental Agriculture (CEA) and District Heating and Cooling (DHC) systems that connect users.

The first option is the simplest, as it could use the larger grid to store and redistribute the additional energy; however, it would disconnect the supply and use of energy locally. The other three options require some form of local energy storage system (ESS) to manage the redistribution. The cost of the ESS is based on its ability to handle peak capacity generation; UMI calculates what the system would need to cope with peak daily and hourly outputs of 21,855 and 911 kW, respectively. Based on the estimated recharge rate and storage capacity, the notional cost of the system would be about €4.5 m, based on a value of €200 per kWh (https://www.nrel.gov/docs/fy19osti/73222.pdf) (accessed on 21 February 2021). Ideally, the best option for the local use of this stored energy depends on matching the temporal patterns of energy generation and use. The correspondence between PV energy output and office cooling demand match closely, as both peak in the warm season, but the maximum cooling demand in July is 370 kWh, which is 380 kWh lower than PV energy available. By comparison, the LUAS system has a near constant monthly energy demand of 357 MWh, such that from March to September there is a surplus (peaking in July) and there is a deficit in the winter half of the year (https://www.tii.ie/public-transport/luas/track-power/) (accessed on 21 February 2021). Finally, the solar energy resource is not a good match to heating energy demand, as the peak supply and demand occur at opposite times of the year. The best solution is to use the energy to run the LUAS (reducing dependence on the national grid) and to use extra energy in the summer months to provide office cooling and/or meet residential needs for lighting and water heating. Figure 6 shows a closed loop system that uses the canal as a cooling/heating resource; during summer, canal water temperatures are about 11 °C, or about 5 °C cooler that the average daytime high. Available surplus energy from the PV arrays, having met the LUAS demand, could power the plant during summer peak demands, essentially converting solar energy into cooling loads. A district heating/cooling system with a capacity to manage the estimated peak cooling demand of 307 kW and an annual load of roughly 13,000 MW at €0.07 kWh coupled with the cost of 650 €/m piping, would cost approximately €2.65 m [40].

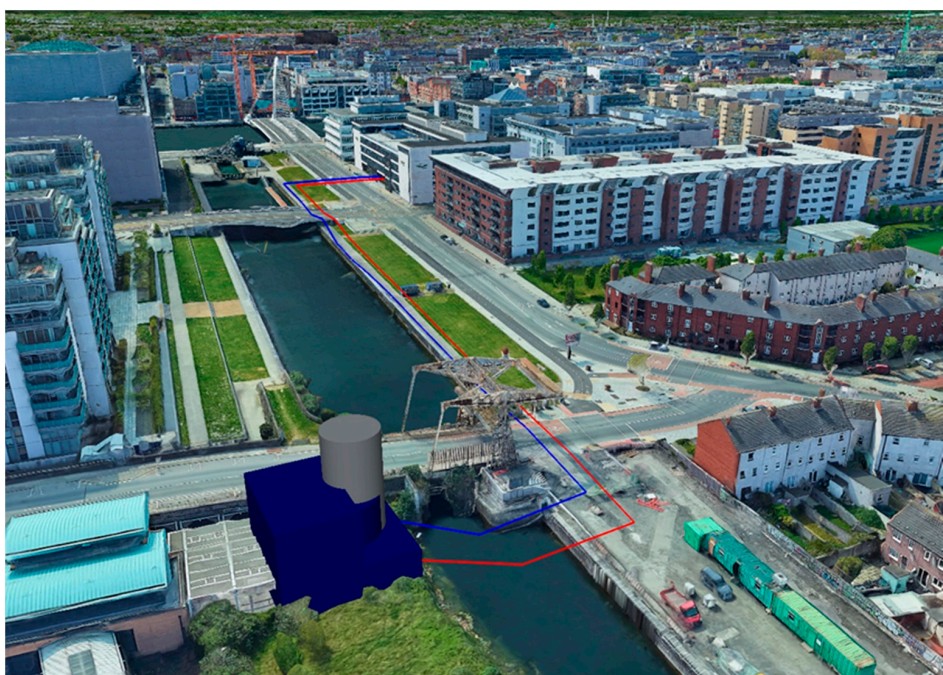

**Figure 6.** Depiction of a district energy plant located on the canal that uses a closed loop system to meet the summertime cooling needs of offices.

## 5. Discussion

The EU Green Deal seeks to reduce $CO_2$ emissions by 55% of 1990 levels by 2030 and to achieve carbon neutrality by 2050. Typically, emissions are categorised by source into residential, commercial, transport, industry and agriculture, each of which is associated with distinct policies. However, many of these functions are spatially 'bundled' together in urban areas where there is the potential for devising policies that are neighbourhood-rather than sector-based. In the case study area selected here, commercial and residential land-uses are co-located with old (canal) and new (transport) transportation systems. An UBEM is ideally suited to looking at neighbourhoods in a holistic manner by reducing energy demand, decarbonising energy supply and managing energy uses.

Here the focus was on housing within the neighbourhood, much of which will remain in place through 2050 and will require significant refurbishment. Table 4 shows the estimated cost of retrofits by type of dwelling. Treating each dwelling as an independent entity results in an estimated cost of about €35 m, but this could be reduced considerably if economies of scale were applied. The most expensive retrofits are needed for the pre-1980 housing stock, which accounts for more than 20% of this budget; coincidentally, these dwellings have a strong link to social deprivation (Table 1). The largest component of the retrofit budget (52%) is allocated to apartment blocks constructed after 2000, simply due to the size of the conditioned area. The impact of the retrofits is to reduce energy demand from this building stock by 88% and the $CO_2$ emissions from 6741 to 839 $tCO_2$. This means that the carbon savings amount to about €5900 per $tCO_2$; this equates to an annual saving of €5 m and a payback period of 7.5 years, based on the current cost of electricity obtained from the national grid (Figure 5). Adding new, energy-efficient stock to the neighbourhood lowers the neighbourhood EUI and, although the total EUI increases, the per capita energy use is reduced by 40% (from 7877 to 4786 kWh per cap).

A neighbourhood perspective can consider energy sharing among different urban functions that are co-located. Here, we have used the LUAS light rail system, which has a constant energy demand through the year (https://www.tii.ie/public-transport/luas/track-power/) (accessed on 21 February 2021) as an ideal way of decoupling the energy demand in the neighbourhood from the national grid. From October to February, the available solar energy cannot meet all the needs of the LUAS, but for the remainder of the

year, there is surplus energy generated. The PV installation described above costs about €20 m for the system, including the ESS, and generates 4865 MWh of energy, which is about the equivalent of 1400 $tCO_2$ saved (or €1.12 m) based on the carbon concentration and cost of electricity grid supply. This equates to nearly €15,000 per $tCO_2$ saved and a payback period of 18 years.

**Table 4.** The estimated cost and impacts of retrofits by dwelling age and type.

| Property | SFH Pre–1980 | SFH 1981–1999 | AB 1981–1999 | AB 2000+ | ND |
|---|---|---|---|---|---|
| Conditioned area (m$^2$) | 31,488 | 16,954 | 31,469 | 75,403 | 118,125 |
| Population | 600 | 350 | 782 | 1879 | 2940 |
| *Status Quo* | | | | | |
| EUI (kWhm$^{-2}$) | 312.9 | 202.9 | 147 | 139.6 | 24.6 |
| EU (MWh) | 985 | 3440 | 4626 | 1053 | 2906 |
| *Retrofit* | | | | | |
| EUI (kWhm$^{-2}$) | 23 | 22.1 | 21.7 | 23.3 | 24.6 |
| EU (MWh) | 724 | 375 | 683 | 1757 | 2906 |
| Change % | 92.6 | 89.1 | 85.2 | 83.3 | 0 |
| *Costs* | | | | | |
| Cost (€m$^{-2}$) | 231 | 117 | 239 | 239 | 0 |
| Cost (€) | 7,273,728 | 1,983,618 | 7,521,091 | 18,021,317 | 0 |
| Cost (€/cap) | 12,123 | 5667 | 9618 | 9591 | 0 |

There are other methods that can be used to offset local drivers of carbon emissions, such as that associated with the supply of food [29]. UMI allows assessment of controlled environmental agriculture (CEA) using the Harvest plugin, which uses electricity to power UV lighting and climate control systems in a space that is isolated from the outdoor climate. The closed environment regulates nutrients, energy and water to optimise productivity and allows multiple layers of crops to be stacked on the same floor space. It also provides insights on the economic performance of the farms through metrics such as operational costs and jobs created locally. Additionally, it compares simulation outputs to existing urban supply chains and provides a carbon balance as well as a site profit. However, unlike the energy harvesting options discussed, the CEA is aspatial in that it can be located in any part of the city and is not tied to geographically-specific resources. Here we used the model to examine the potential for growing tomatoes, lettuce and broccoli, which are relatively high value crops currently imported from Mediterranean climates (distance of 4500 km). If we consider a future city in which fewer private cars are used, then some existing car spaces can be re-purposed.

In the case examined here, three floors of an existing multi-storey carpark (Figure 1) with a total area of 5673 m$^2$ are converted into a CEA farm. The energy and water needs could be met from a combination of conventional and local energy and water supplies. Based on price, local consumption, water and energy demand, and production-related $CO_2$ emissions, UMI estimates the carbon savings and profit. We estimate that such a facility would have an energy demand of 2229 MWh (equivalent 745 $tCO_2$ and €580,000 from the national electricity grid) (Figure 7). The output of the CEA would satisfy 28%, 4% and 1% of Dublin's lettuce, broccoli and tomato demand, respectively (http://www.fao.org/faostat/en/#data) (accessed on 1 March 2021). The estimated $CO_2$ savings associated arising from substituting CEA products for those imported is 5885 $tCO_2$, based on FAO values. The cost of the CEA infrastructure varies between €7.5 m

and €12.5 m, which equates to nearly €1200–2200 per tCO$_2$ saved. The payback period, using the national electricity supply as the benchmark, is less than 3 years. There are other costs involved; the estimated annual nutrient bill is €300,000 and the water cost is €25,000, although the latter could draw on the canal for supplies.

**Area: 5673 m² over three floors**
**Energy demand: 2229455 kWh**
**Water demand: 19,203 m³**

**Lettuce**
Energy: 585 kWh/m²/year
Water: 7.99 m³/m²/year

**Tomatoes**
Energy: 222 kWh/m²/year
Water: 0.56 m³/m²/year

**Broccoli**
Energy: 372 kWh/m²/year
Water: 1.61 m³/m²/year

**Figure 7.** A closed agricultural environment (CEA) and the inputs needed to grow lettuce, tomatoes and broccoli. The inputs needed for a CEA that occupies three floors of a parking structure are listed. The photograph shows lettuce plants being grown in a vertically-stacked hydroponic system (VertiCrop System by Valcenteu, licensed under CC by 3.0).

The best mix of policy strategies to achieve the EU's targets under the Green Deal would focus on building retrofits as the best way to reduce energy demand to meet heating needs. In any scenario, improving building facades in the study has a greater impact than any other initiative, and the advantage of the UBEM is to identify clusters of buildings where multi-dwelling retrofits can reduce overall costs. The addition of a large new development that is energy efficient increases the total neighbourhood demand but reduces the energy intensity (by area and by population). The generation and sharing of PV energy locally can offset some of the residential energy needs but is probably best used to support the electrified mass transit system (LUAS) and summer cooling needs of offices using the nearby canal. The investment and payback period for this policy is longer but has the advantage of linking land-uses locally and reducing reliance on outside energy imports. In the case of the LUAS, the neighbourhood would get carbon credit for replacing the carbon emissions of commuters. Finally, the addition of CEA is a relatively inexpensive way of offsetting indirect carbon emissions associated with food production and transport but, like the LUAS, the carbon benefits extend well outside the neighbourhood.

## 6. Conclusions

The study presented here shows the value of an area- rather than sector-based approach to achieving energy resilience in a neighbourhood. This approach requires the appropriate tools and data to assess the diversity of energy demands associated with building types and occupancy patterns. Urban building energy models (UBEMs) are ideally suited to this task, as they can simulate energy demand across urban landscapes and can incorporate novel approaches to managing energy use and generating and sharing energy locally. However, the software and data requirements of these models can be onerous.

Here, we used the publicly available Urban Modelling Interface (UMI) to test the best fit carbon reducing policies in a complex urban geography, given the context of the EU Green Deal and the short timeline for meeting its goals. This is the first application of this model to neighbourhood-scale energy management, using Tabula building archetype data, which is available for over 20 countries in Europe. While the work here focused on

one neighbourhood in Dublin (Ireland) using the national archetypes, the broader work project has generated UMI building templates for all Tabula archetypes. The remaining challenge is to link these archetypes to their geography using building footprints, so that the work shown here can be replicated. This research shows the potential for UBEM to create bespoke energy policies for complex neighbourhoods that can account for different land-covers and land-uses and their time varying energy needs. The path shown in this paper is an efficient way of applying and testing these models.

**Author Contributions:** Conceptualisation; methodology; validation; formal analysis; investigation; resources; data curation; writing—original draft preparation, N.B. Writing—review and editing; visualisation; G.M., S.L.-D., K.B. and N.B.; supervision; project administration; funding acquisition, G.M. Software; S.L.-D. and K.B. All authors have read and agreed to the published version of the manuscript.

**Funding:** This work has been funded by the Sustainable Energy Authority of Ireland under the SEAI Research, Development and Demonstration Funding Program 2018, Grant number 18/RDD/232.

**Institutional Review Board Statement:** Not applicable.

**Informed Consent Statement:** Not applicable.

**Conflicts of Interest:** The authors declare no conflict of interest.

## Appendix A

| Controlled Environmental Agriculture | | |
|---|---|---|
| **Description** | **Units** | **Source** |
| Yearly Demand for Broccoli, tomatoes, and lettuce | kg/cap/year | http://www.fao.org/faostat/en/#data |
| Workers Density | Worker/ m² | https://www.freightfarms.com/build-your-business |
| Workers Wage | €/year | https://www.payscale.com/research/IE/J |
| Water footprint (at source) | L/kg | https://waterfootprint.org/ |
| Emissions Factor (at source) | kgCO2eq/wKh | https://www.sciencedirect.com/Êscience/article/pii/S1361920916307933 |
| Emissions Factor (on site) | kgCO2eq/wKh | https://www.seai.ie/data-and-insights/seai-statistics/conversion-factors/ |
| Cost of Electricity (on site) | €/kWh | https://www.seai.ie/data-and-insights/seai-statistics/key-statistics/prices/ |
| **District Cooling Plant** | | |
| **Description** | **Units** | **Source** |
| Plant Price | €/kWh | Swedblom *et al.*, 2015 |
| Piping Price | €/meter | Swedblom *et al.*, 2015 |
| **Energy Storage System** | | |
| **Description** | **Units** | **Source** |
| Battery Cost | €/kWh | https://www.nrel.gov/docs/fy19osti/73222.pdf |
| **Luas** | | |
| **Description** | **Units** | **Source** |
| Energy Needs | kWh/ per journey | Sweeney, 2015 |
| Power Supply | kV AC | https://www.tii.ie/public-transport/luas/track-power/ |
| **PV** | | |
| **Description** | **Units** | **Source** |
| Cost of PV installations | €/m² | https://www.seai.ie/publications/Best_Practice_Guide_for_PV.pdf |

**Figure A1.** Sources used for estimating cost-, energy- and $CO_2$-related savings are listed in the Appendix of this paper.

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
