# Peer review of "Designing an Energy-Resilient Neighbourhood Using an Urban Building Energy Model"

_energies, doi:10.3390/en14154445_

Round 1
Reviewer 1 Report
This is an interesting paper on the subject of urban building energy modelling. The paper can be improved following the revisions suggested below:
- A thorough proof reading before final submission: the quality of writing and the presentation and communication of the paper is generally very good. However, the manuscript will benefit from another round of proof reading (e.g. a word is probably missing in line 95, line 109: the main obstacle 'against' their wider use, lines 80-82 could also be reworded to make it more understandable for prospective readers.)
- No work is presented for the validation of the urban building energy model. Whilst there is reference to previous work covering the residential buildings, office buildings are also considered in this study for energy harvesting and sharing energy resources. It is necessary to address model validation more explicitly in the text, especially with regards to the non-domestic buildings included in the model.
- Is Figure 4 reflective of 2020 or 2050? Figure 3 compares heating degree-days for 2020 and 2050. It would be helpful to evaluate the changes in heating/cooling demand also between 2020 and 2050, or at least provide further clarification about Figure 4.
- Table 3: what would be the impact of these measures on energy use? It would be helpful to quantify the impact of fabric measures and changes in building services (shift to heat pump), and lighting improvements separately along with associated costs. It is stated in the discussion section that fabric measures are the most effective measures (lines 469-470). It would be helpful to quantify and provide the effect of these measures on buildings' energy demand separately.
- What was the source for cost estimations in Table 3?
- A large part of your discussion section is allocated to testing a new idea (CEA) and associated energy demand and emission saving calculations. This could be integrated into methodology and result sections with a clearer presentation of the underlying assumptions used to derive the provided estimations.
- What is the contribution to knowledge here? What makes your paper distinctive from a consultancy report? It would be helpful to try and identify key contributions made to the knowledge in this field (e.g. any methodological novelty used in developing the urban building model) and clearly highlight it in the paper.
Author Response
Please find the response attached.

Reviewer 2 Report
The paper presents the application of UBEM to a case study of a Dublin neighborhood. The authors analyze the building stock through UMI (Urban Modelling Interface), defining archetype buildings, on which they perform energy analysis in the current state and with the hypothesis of energy retrofit.
The paper presents an argument concerning an interesting topic, but it cannot be published in the present form.
The main comments are summarized as follows:
- Title and abstract are clear. However, I suggest adding some of the more significant findings, possibly numerical, in the abstract.
- I recommend re-reading the article to eliminate typos (e.g., line 34, line 45, etc.) and some grammatical errors.
- When acronyms are introduced, an explanation should be given immediately (e.g., line 166).
- Figure 1. The age classes shown in the legend range from 1900-1930 to 1980. So, no buildings were constructed between 1930 and 1980?
- Table 1 is not easy to understand:
- What do the codes in the first column represent? For what reason were they included?
- What does the acronym "NRG" mean?
- What do the codes associated with the medians of the BERs mean? If they are performance classes, they should be made explicit.
- What is the meaning of the Surveys in the last column? How were they generated?
- The “Deprivetion Index” concept is not very clear. I recommend expanding the explanation to make it easier to understand and useful.
- Regarding the "age of housing", in Table 1 the authors group the buildings into three classes (pre-1980, 1981-1999, 2000+), with the pre-1980 class being the most relevant. In my opinion, including the 304 buildings in a single class could be an oversimplification, because it means that buildings built in 1900-1930 (for example) have the same characteristics as those built in 1980. So, I suggest expanding the age classes in the table.
- Figures should be referred to and described in the text before being shown. For example, figure 2.
- It is necessary to check the numbering of the figures. In some cases (e.g., figure 2) the wrong figure is indicated.
- The description of the epw files generation for future climate conditions is overly brief. Technically, what approach was used to generate the climate change files?
- Figure 4. If the curve relative to PV is energy generation, why does the y-axis show the "energy demand"? I would suggest modifying the figure introducing an additional axis.
- Regarding the simulated heating and cooling demand, what can the authors say about the model calibration/validation process?
- The numbering of the tables is incorrect and makes it difficult to understand the text.
- Table 3. Authors should detail how costs were defined.
- Figure 5. Please, explain the meaning of the acronyms.
- Line 356. What does €17m mean? What unit of measurement is it?
Author Response
Please find the response attached.

Reviewer 3 Report
The authors have presented a detailed study that adds to the body of knowledge. However, the following can be improved:
Clearly stated research questions/aims/objectives that connects the methodology
The literature review section can be improved to focus on the topic
The methodology section should clearly tie the research questions
The discussions section should relate closely with the analysis and new findings.
Author Response
Please find the response attached.

Round 2
Reviewer 2 Report
I appreciated the authors' effort in following up on my comments/suggestions. In my opinion, the revised version of the manuscript can be accepted for publication.